# Complex pain phenotypes: Suicidal ideation and attempt through latent multimorbidity

**Kangwon Song**[1], **Ben J. Brintz**[2], **Chen-Pin Wang**[3], **Donald D. McGeary**[4], **Cindy A. McGeary**[4], **Jennifer S. Potter**[4], **Carlos A. Jaramillo**[5], **Blessen C. Eapen**[6,7], **Mary Jo Pugh**[2,8]*

1 Department of Pharmacy, AU Medical Center, Augusta University, Augusta, GA, United States of America, 2 Division of Epidemiology, Department of Internal Medicine, University of Utah, Salt Lake City, UT, United States of America, 3 Department of Population Health Sciences, University of Texas Health Science Center San Antonio, San Antonio, TX, United States of America, 4 Department of Psychiatry and Behavioral Sciences, University of Texas Health Science Center San Antonio, San Antonio, TX, United States of America, 5 Department of Physical Medicine and Rehabilitation, South Texas Veterans Health Care System, San Antonio, TX, United States of America, 6 Department of Physical Medicine and Rehabilitation, VA Greater Los Angeles Health Care System, Los Angeles, CA, United States of America, 7 Division of Physical Medicine and Rehabilitation, Department of Medicine, David Geffen School of Medicine at UCLA, Los Angeles, CA, United States of America, 8 VA Salt Lake City Health Care System, IDEAS Center, Salt Lake City, UT, United States of America

☯ These authors contributed equally to this work.
* Maryjo.pugh2@va.gov

**Data Availability Statement:** Data are owned by the VA and it is not possible to provide de-identified VA health system data of Veterans outside the VA firewall. In this instance it is even more problematic

## Abstract

### Background

Given the relatively high rates of suicidal ideation and attempt among people with chronic pain, there is a need to understand the underlying factors to target suicide prevention efforts. To date, no study has examined the association between pain phenotypes and suicide related behaviors among those with mild traumatic brain injuries.

### Objective

To determine if pain phenotypes were independently associated with suicidal ideation / attempt or if comorbidities within the pain phenotypes account for the association between pain phenotypes and suicide related behaviors.

### Methods

This is a longitudinal retrospective cohort study of suicide ideation/attempts among pain phenotypes previously derived using general mixture latent variable models of the joint distribution of repeated measures of pain scores and pain medications/treatment. We used national VA inpatient, outpatient, and pharmacy data files for Post-9/11 Veterans with mild traumatic injury who entered VA care between fiscal years (FY) 2007 and 2009. We considered a counterfactual causal modeling framework to assess the extent that the pain phenotypes during years 1–5 of VA care were predictive of suicide ideation/attempt during years 6–8 of VA care conditioned on covariates being balanced between pain phenotypes.

as data include information on substance use disorder, suicidal ideation and suicide attempt. This is based on the VA Research and Development Committee, VA Salt Lake City (Caroline. phinney@va.gov) restrictions. These data are available to interested parties who complete the requirements to access the data behind the VA firewall.

**Funding:** MJP. R21 HD089098-01. The National Institute of Child Health and Human Development. https://www.nichd.nih.gov/ The funders had no role in study design, data collection and analysis, decision to publish, or preparation of the manuscript. MJP. IK6HX002608. VA Health Services Research and Development Service. https://www.research.va.gov/services/hsrd.cfm The funders had no role in study design, data collection and analysis, decision to publish, or preparation of the manuscript. BJB. 8UL1TR000105 (formerly UL1RR025764). National Center for Research Resources and the National Center for Advancing Translational Sciences, National Institutes of Health. https:// ncats.nih.gov/ The funders had no role in study design, data collection and analysis, decision to publish, or preparation of the manuscript. For the remaining authors no funding sources were declared.

**Competing interests:** The authors have declared that no competing interests exist. Any opinions, findings, conclusions, or recommendations expressed in this publication are those of the authors and do not necessarily reflect the views of the U.S. Government, or the U.S. Department of Veterans Affairs, and no official endorsement should be inferred.

## Results

Without adjustment, pain phenotypes were significant predictors of suicide related behaviors. When we used propensity scores to balance the comorbidities present in the pain phenotypes, the pain phenotypes were no longer significantly associated with suicide related behaviors.

## Conclusion

These findings suggest that suicide ideation/attempt is associated with pain trajectories primarily through latent multimorbidity. Therefore, it is critical to identify and manage comorbidities (e.g., depression, post-traumatic stress disorder) to prevent tragic outcomes associated with suicide related behaviors throughout the course of chronic pain and mild traumatic brain injury management.

## Introduction

Mild traumatic injury (mTBI) is a signature injury of war in Iraq and Afghanistan Veterans [1, 2]. Although, the mechanism is unknown, mTBI is associated with up to a two-fold increased risk of suicide [3]. Following mTBI, musculoskeletal and headache pain is commonly reported. Pain is a factor that may interact with mTBI to increase the risk of suicide.

Some individuals with chronic pain consider suicide as an option to eliminate suffering and burden to others when pharmacological, medical, and surgical therapies are exhausted [4–9]. Suicidal ideation, attempts and completion are two to three times more likely in those with chronic pain than in those without pain [10–13]. The intersection between longitudinal patterns of pain, mTBI and suicide presents clear challenges to the patients and providers. However, prior research examining longitudinal patterns of pain trajectories, multifaceted treatment, and subsequent adverse outcomes such as suicidal ideation and attempt (suicide-related behavior; SRB) are limited among patients with mTBI [14].

Our prior work identified pain trajectory phenotypes in Post-9/11 Veterans with mTBI using latent trajectory models. We found that four pain trajectory phenotypes (complex low impact, stable pain; complex low impact, worsening pain; complex moderate impact, worsening pain; complex high impact, stable pain) characterized by different patterns of longitudinal pain trajectories and complex, multidimensional longitudinal treatment regimens including medications with dual indications for use (e.g., antidepressants, anticonvulsants) [15]. Because prior research in Post-9/11 Veterans has also found that chronic pain frequently occurs with mTBI, post-traumatic stress disorder (PTSD), and depression [16, 17], our examination includes not only the pain trajectory phenotypes (hereafter pain phenotypes), but also comorbid conditions that may be reflected in the complex treatment regimens reflected in the pain phenotypes. These comorbid conditions account for the mental pain and suffering that can contribute to suicide risk via defeatist thoughts and emotions that color future expectations and threaten a positive future [18].

Different components of complex pain may have additive effects on suicide ideation and attempt including medications, high pain scores, and accumulated non-cancer pain conditions [5, 19–23]. Complex comorbidity may add to the impact of these components of complex pain on SRB [24–26], or they may account for the relationship between pain phenotypes and SRB among Veterans with mTBI [3, 27–33]. We sought to determine if these pain phenotypes in

mTBI were independently associated with SRB, or if the complexity of polymorbidity that may be a latent component of the pain phenotypes accounts for the association between pain phenotypes and SRB.

## Materials and methods

### Study cohort

The study sample and methods for development of pain phenotypes are described previously [15]. This study's analytic sample included those who entered VA care October 1, 2007 through September 30, 2009, who were diagnosed with mild TBI (mTBI), and who had at least three years of care during the first 5 years after entering VA care [34]. The University of Texas Health Science Center at San Antonio, the University of Utah, and the Bedford VHA Hospital institutional review boards approved the study, with a waiver of informed consent.

### Measures and data sources

**Primary outcome: Suicide related behavior.** We used ICD9-CM codes used in prior studies of SRB to identify suicide ideation (V6284) and attempt (E950, E952, E953, E953, E954, E955, E956, E957, E958, E959) in national VA inpatient and outpatient data in years 6–8 of VA care [35]. Due to small numbers of documented attempts, we examined three SRB categories: suicidal ideation without attempt, suicide attempt regardless of suicide ideation, and no SRB. We also examined the combined outcome of any SRB vs. no SRB.

**Primary independent variable: Pain phenotype.** As described in previous work by Song, et al. [15], pain phenotypes were derived from general mixture latent variable models based on repeated measures of pain scores and pain medications or other pain treatments (e.g., complex pain clinic, physical therapy) during years 1–5 of VA care. These models identified four pain phenotypes: 'low impact-stable,' 'low impact- worsening,' 'moderate impact-worsening,' and 'high impact-stable.'

**Covariates.** We included age, sex, race/ethnicity, education, and military characteristics from the Operation Enduring Freedom/Operation Iraqi Freedom/ Operation New Dawn Roster file through December 2014, and percent service-connected disability from Veteran's Service Network data. Age was classified as 18–29, 30–39, 40–49 and greater than or equal to 50 years. Sex was classified as male and female based on data reported from the Department of Defense presented at the time of entry to military service. Race/ethnicity was defined as Black, White, Hispanic, or other based on available numbers in each racial/ethnic group. Education was the education level at the time of leaving military service classified as high school or less and some post-high school education and higher. Military characteristics included branch of service (Army, Marine Corps, Navy/Coast Guard, and Air Force), component of service (Active Duty, National Guard/Reserve), military rank (enlisted, officer/warrant officer), and deployment history (single deployment, multiple deployments). Service-connected disability is awarded in 10% increments and was classified as 0–20%, 30–50%, 60–80%, and 90–100%.

We identified diagnosed health conditions hypothesized to be reflected in the pain phenotypes and associated with SRB [5, 10, 24–26, 36]. S1 Table provides ICD9-CM diagnosis codes used; the algorithm required two or more diagnoses at least 7 days apart in outpatient data or at least once in inpatient data during the period of pain trajectory class analysis (years 1–5 of VA care) [37, 38]. Conditions included as covariates were back/neck pain, other musculoskeletal pain, headache, posttraumatic stress disorder (PTSD), depression, anxiety, substance use disorder, insomnia, attention impairment (proxy for potential impulsiveness), and cognitive dysfunction.

## Statistical analysis

Descriptive statistics for each pain phenotype were calculated. Chi-square tests were used to compare categorical variables between pain phenotypes.

We considered a counterfactual causal modeling framework [39, 40] to assess the extent that the pain phenotypes during years 1–5 of VA care were predictive of SRB during years 6–8 of VA care should covariates be balanced between pain phenotypes. The propensity scores of pain phenotypes were calculated, and the inverse propensity scores were incorporated as weights in the SRB model to minimize observed confounding for assessing the association of pain phenotype with SRB. To obtain robust estimates of propensity scores associated with pain phenotypes, we used generalized boosted multinomial regression modeling implemented by the twang package in R [41]. To examine the extent to which comorbidities contribute to differential SRB by pain phenotype, we considered two propensity score models with nested sets of predictors. Set 1 (long set) predictors were age, sex, race/ethnicity, education, military rank, multiple deployments, service-connected disability, prior SRB, and diagnosed health conditions that include headache, pain, anxiety, depression, PTSD, insomnia, substance use disorder, attention impairment and cognitive dysfunction. Set 2 (short set) predictors were a subset of Set 1 covariates excluding comorbid health conditions. Propensity score models were evaluated in terms of covariate balance between pain phenotypes using standardized bias—a standardized bias below the threshold 0.25 was deemed as covariate balance [42].

For SRB, we considered models for both a dichotomous and a trichotomous outcome. For any SRB or none, we conducted three separate logistic regression models to estimate the effects of pain phenotypes in SRB conditioned on varying degrees of adjustment of confounding associated with covariates, including (1) all covariates (long set) as predictors and inverse propensity score weights (IPSW) predicted by all covariates; (2) the short list of covariates (that excludes comorbid health conditions) as predictors and IPSW predicted by the short list of covariates; and (3) none. For the trichotomous SRB outcome (any attempt, ideation only, or no SRB), three multinomial logistic regression analyses were conducted with adjustments (1)-(3) as described above.

In each adjusted model of SRB, we included all predictors in the propensity score model (short or long set) as predictors along with the inverse propensity score weights adjustment so that the estimates associated with pain phenotypes have the 'double-robustness' property, i.e., the estimator is consistent either the propensity score model or the outcome model is correctly specified [43]. Under the four assumptions of consistency, exchangeability, positivity, and no misspecification of both outcome and propensity score models, the IPSW estimates of ORs associated with pain phenotypes based on adjustment (1) are interpreted as the effects of pain phenotypes conditioned on the counterfactual all covariates being balanced among phenotypes [44]. Similarly, the IPSW adjusted OR estimates based on the adjustment (2) are interpreted as the effects of pain phenotype on SRB should the partial set of covariates be balanced among pain phenotypes. Comparing estimates associated with pain phenotypes between these two IPSW adjustments will allow us to assess the extent to which the comorbid health conditions jointly attributed to the differential SRB risks between pain phenotypes.

## Results

All 10,717 Veterans with mTBI had complex pain and multiple pain treatment modalities. Proportions of Veterans by phenotype were: 'low impact-stable' 33.8%, 'low impact- worsening' 19.1%, 'moderate impact-worsening' 33.2%, and 'high impact-stable' 18.7%. The 'high impact-stable' phenotype had the highest prevalence of SRB followed by the 'low impact-worsening' phenotype. Table 1 shows that pain phenotype was associated with well-known SRB

**Table 1. Demographic and clinical characteristics by complex pain phenotype.**

| Characteristics | Complex pain phenotype, N = 10717 | | | | | | | | |
|---|---|---|---|---|---|---|---|---|---|
| | Low impact, stable | | Low impact, worsening | | Moderate impact, worsening | | High impact, stable | | |
| | n = 3454 (33.8%) | | n = 1955 (19.1%) | | n = 3393 (33.2%) | | n = 1915 (18.7%) | | P value |
| Age: ≤29 | 2479 | 72% | 1289 | 66% | 2136 | 63% | 1088 | 57% | < .001 |
| 30–39 | 548 | 16% | 429 | 22% | 700 | 21% | 517 | 27% | |
| 40–49 | 342 | 10% | 190 | 10% | 469 | 14% | 269 | 14% | |
| 50+ | 85 | 2% | 47 | 2% | 88 | 3% | 41 | 2% | |
| Sex: Male | 3226 | 93% | 1828 | 94% | 3184 | 94% | 1779 | 93% | .61 |
| Female | 228 | 7% | 127 | 6% | 209 | 6% | 136 | 7% | |
| Race: Black | 437 | 13% | 204 | 10% | 530 | 16% | 199 | 10% | < .001 |
| White | 2430 | 70% | 1400 | 72% | 2274 | 67% | 1393 | 73% | |
| Hispanic | 433 | 13% | 263 | 13% | 446 | 13% | 236 | 12% | |
| Other | 154 | 4% | 88 | 5% | 143 | 4% | 87 | 5% | |
| Education: High school or less | 3055 | 88% | 1684 | 86% | 3004 | 89% | 1679 | 88% | .04 |
| Some college plus | 399 | 12% | 271 | 14% | 389 | 11% | 236 | 12% | |
| Rank: Enlisted | 3346 | 97% | 1891 | 97% | 3322 | 98% | 1872 | 98% | .01 |
| Officer/Warrant | 108 | 3% | 64 | 3% | 71 | 2% | 43 | 2% | |
| Service Branch: Army | 2237 | 65% | 1407 | 72% | 2431 | 72% | 1443 | 75% | < .001 |
| Air Force | 130 | 4% | 68 | 3% | 138 | 4% | 86 | 4% | |
| Navy/Coast Guard | 240 | 7% | 141 | 7% | 245 | 7% | 133 | 7% | |
| Marines | 847 | 25% | 339 | 17% | 579 | 17% | 253 | 13% | |
| Component: Active | 2484 | 72% | 1363 | 70% | 2423 | 71% | 1340 | 70% | .24 |
| Reserve/National Guard | 970 | 28% | 592 | 30% | 970 | 29% | 575 | 30% | |
| Multiple Deployments | 1905 | 55% | 976 | 50% | 1627 | 48% | 824 | 43% | < .001 |
| Service Connected Disability: 0–20% | 646 | 19% | 187 | 10% | 380 | 11% | 163 | 9% | < .001 |
| 30–50% | 670 | 19% | 192 | 10% | 373 | 11% | 73 | 4% | |
| 60–80% | 1400 | 41% | 763 | 39% | 1326 | 39% | 556 | 29% | |
| 90–100% | 738 | 21% | 813 | 42% | 1314 | 39% | 1123 | 59% | |
| Suicidal Ideation/Attempt, Y1-5: none | 3262 | 94% | 1679 | 86% | 3108 | 92% | 1573 | 82% | < .001 |
| Suicide Attempt | 28 | 1% | 30 | 2% | 29 | 1% | 38 | 2% | |
| Suicidal Ideation | 145 | 4% | 208 | 11% | 214 | 6% | 229 | 12% | |
| Suicidal Ideation + Suicide Attempt | 19 | 1% | 38 | 2% | 42 | 1% | 75 | 4% | |
| Comorbidities, Y1-5[a] | | | | | | | | | |
| Headache | 1327 | 38% | 1071 | 55% | 1968 | 58% | 1337 | 70% | < .001 |
| Back/neck pain | 1700 | 49% | 1298 | 66% | 2776 | 82% | 1746 | 91% | < .001 |
| Other musculoskeletal pain | 1507 | 44% | 1126 | 58% | 2343 | 69% | 1628 | 85% | < .001 |
| Anxiety | 993 | 29% | 882 | 45% | 1107 | 33% | 867 | 45% | < .001 |
| Depression | 1608 | 47% | 1386 | 71% | 2025 | 60% | 1477 | 77% | < .001 |
| Post-traumatic stress disorder | 2534 | 73% | 1820 | 93% | 2846 | 84% | 1821 | 95% | < .001 |
| Insomnia | 645 | 19% | 800 | 41% | 812 | 24% | 784 | 41% | < .001 |
| Substance use disorder | 1071 | 31% | 767 | 39% | 1126 | 33% | 874 | 46% | < .001 |
| Attention impairment | 185 | 5% | 174 | 9% | 225 | 7% | 194 | 10% | < .001 |
| Cognitive dysfunction | 126 | 4% | 110 | 6% | 173 | 5% | 132 | 7% | < .001 |

[a] Additional comorbidity variables in the long model

predictors: younger age, race/ethnicity, education, rank, service branch, multiple deployments, service-connected disability, pain and mental health conditions, and prior suicidal ideation

and attempts. The propensity score model for pain phenotypes predicted by the long set of covariates achieved balances of all covariates between pain phenotypes (standardized biases associated covariates all fell below the 0.25 threshold), but the propensity score model predicted by the short set of covariates did not necessarily balance the health covariates. See S2 Table for binomial logistic regression results and S3 and S4 Tables for the multinomial logistic regression results.

## Modeling any SRB as a dichotomous outcome

Overall, 95% veterans had no SRB and 5% veterans had SRB (Table 2). In the unadjusted binomial logistic regression, 'low impact-worsening' pain, 'moderate impact-worsening' and 'high impact-stable' pain phenotypes were all associated with increased SRB compared to the 'low impact-stable' pain phenotype (Fig 1). After adjusting for demographics and prior SRB among pain phenotypes (IPSW estimates of the effects of pain phenotypes derived from the propensity scores predicted by the short set of covariates), the 'low impact-worsening' and 'high impact-stable' pain phenotypes remained significantly associated with increased risk of SRB. The magnitudes of the odds ratios associated with pain phenotypes were attenuated compared to those in the unadjusted model. After balancing diagnosed health conditions between pain phenotypes (IPSW estimates of the effects of pain phenotypes derived from the propensity scores predicted by the long set of covariates), pain phenotypes were no longer associated with SRB risk.

## Modeling no SRB, suicidal ideation, and suicidal attempt as a trichotomous outcome

For the trichotomous outcome of no SRB, suicidal ideation only, and suicide attempt contained 9751, 390, and 93 veterans, respectively (Table 2).

In unadjusted multinomial logistic regression models, the 'low impact-stable' pain phenotype had significantly lower odds of suicidal ideation compared to other pain phenotypes (Fig 2). The 'low impact-stable' pain phenotype had significantly lower odds of suicide attempt-only compared to the 'high impact-stable' pain. After adjusting for demographic covariates and prior SRB among pain phenotypes (IPSW estimates of the effects of pain phenotypes derived from the propensity scores predicted by the short set of covariates), the 'high impact-stable' pain and 'low impact-worsening' phenotypes remained associated with increased risk for suicidal ideation but with a lesser magnitude compared to the unadjusted model. The 'moderate impact-worsening' pain phenotype was no longer associated with the risk of suicide ideation conditioned on balanced demographic covariates, while the effects of high impact and low impact

**Table 2. Dichotomous and trichotomous analysis suicidal ideation and/or attempt by complex pain phenotype.**

| | Complex pain phenotype, No. (%) | | | | | | | |
|---|---|---|---|---|---|---|---|---|
| | Low impact, stable | | Low impact, worsening | | Moderate impact, worsening | | High impact, stable | |
| **Outcome by analysis** | **n = 3454 (33.8%)** | | **n = 1955 (19.1%)** | | **n = 3393 (33.2%)** | | **n = 1915 (18.7%)** | |
| No suicidal ideation/attempt, n = 9751(95.3%) | 3360 | 97.3% | 1846 | 94.4% | 3272 | 96.4% | 1756 | 91.7% |
| Dichotomous analysis | | | | | | | | |
| Suicidal ideation/attempt, n = 483 (5%) | 94 | 2.7% | 109 | 5.6% | 121 | 3.6% | 159 | 8.3% |
| Trichotomous analysis | | | | | | | | |
| Suicidal ideation, n = 390 (3.8%) | 74 | 2.1% | 92 | 4.7% | 99 | 2.9% | 125 | 6.5% |
| Suicidal attempt, n = 90 (0.9%) | 20 | 0.6% | 17 | 0.9% | 22 | 0.7% | 34 | 1.8% |

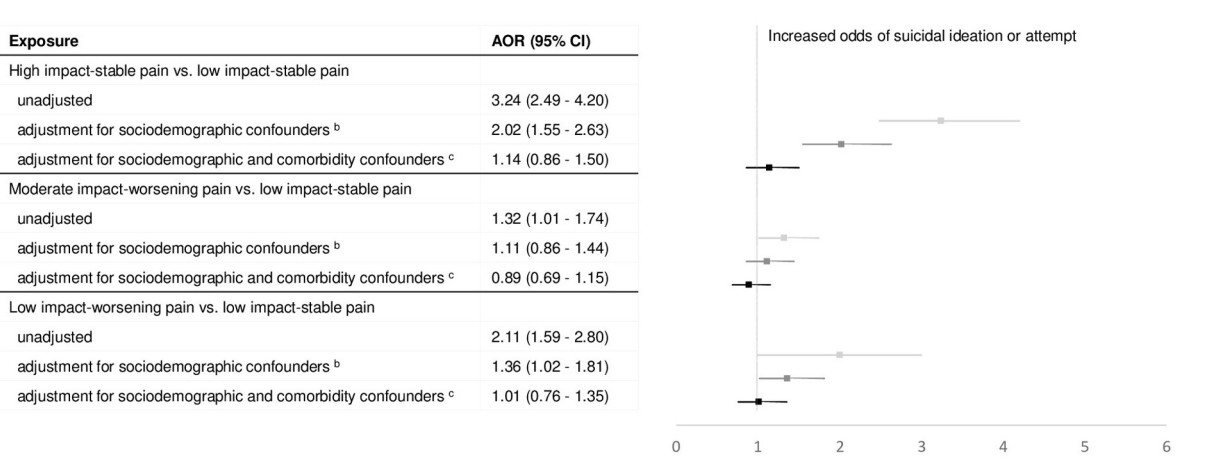

**Fig 1. Effects of pain phenotypes on suicidal ideation or attempt[a].** [a] Forest plot of odds ratios and 95% confidence intervals associated with pain phenotypes derived from logistic regression analyses: unadjusted, adjusting away confounding associated with short set of covariates using IPSW, and adjusting away confounding associated with long set of covariates using IPSW. [b] The short set covariates includes sociodemographic characteristics (excludes comorbid health conditions) as predictors. [c] The long set covariates includes all covariates sociodemographic characteristics and comorbidities.

worsening remained significant but with reduced magnitudes. Pain phenotype was no longer associated with increased risk of suicide attempt in either adjusted models (Fig 3).

## Discussion

To understand the relatively high-risk of SRB in those with complex chronic pain, we examined the association of pain complexity, comorbidities, and suicide. Without adjustment, our

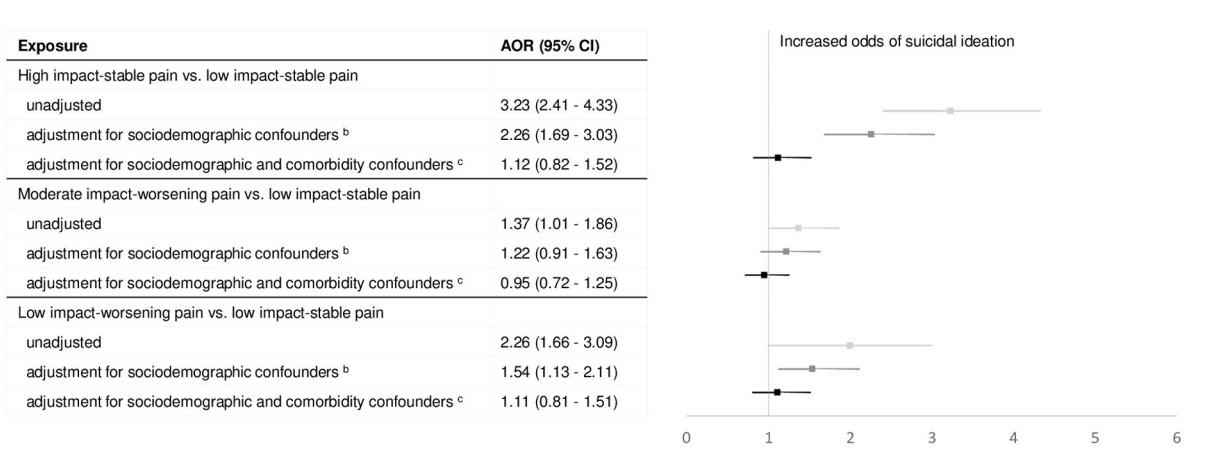

**Fig 2. Effects of pain phenotypes on suicidal ideation.** [a] Forest plot with odds ratio and 95% confidence intervals of the pain phenotypes with low impact-stable pain as reference from the multinomial logistic regression models. The figure shows the effect sizes with and without weights and using the long and short set of covariates for suicidal ideation. [b] The short set covariates includes sociodemographic characteristics (excludes comorbid health conditions) as predictors. [c] The long set covariates includes all covariates–sociodemographic characteristics and comorbidities.

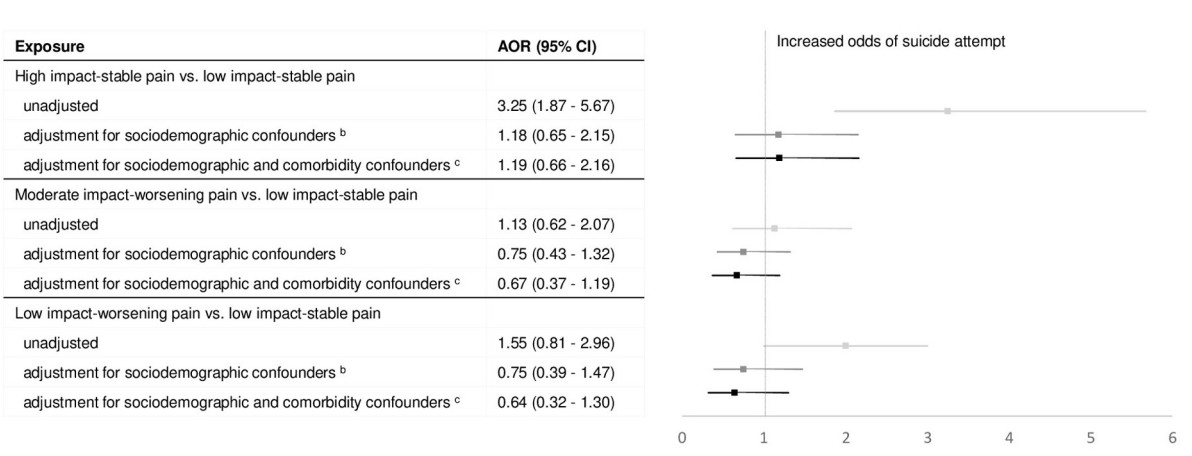

**Fig 3. Effects of pain phenotypes on suicide attempt.** [a] Forest plot with odds ratio and 95% confidence intervals of the pain phenotypes with low impact-stable pain as reference from the multinomial logistic regression models. The figure shows the effect sizes with and without weights and using the long and short set of covariates for suicide attempt. [b] The short set covariates includes sociodemographic characteristics (excludes comorbid health conditions) as predictors. [c] The long set covariates includes all covariates–sociodemographic characteristics and comorbidities.

previously identified pain phenotypes were significant predictors of SRB. However, this assessment did not account for the unobserved presence of comorbidities in this population. Therefore, we utilized propensity scores to balance the comorbidities present in the pain phenotypes, pain intensity and treatment trajectories and found they were no longer significantly associated with SRB. This indicates that comorbidities were a significant factor associating chronic pain with SRB. This further indicates that assessing, treating, and managing comorbid conditions in people living with pain is important to address risk for SRB.

Veterans in the high impact-stable phenotype were most likely to report suicidal ideation and/or attempt suicide. Individuals in this phenotype had the highest rates of PTSD, depression, SUD, and past suicide attempts. Veterans in the low impact-worsening phenotype were the next most likely to endorse SRB and had the second highest rates of PTSD, depression, SUD, and past suicide attempts. Our findings align with prior research on suicide risk in a military polytrauma sample which found that depression and PTSD were significantly associated with suicidal ideation and violent impulses in veterans with chronic pain [36]. Management of these comorbidities may have the most impact on suicide mitigation.

Veterans in the 'high impact-stable' and 'low-impact worsening' phenotypes had the highest rates of previous suicide attempts and SRB after phenotype development. This finding is consistent with the vast majority of studies that find the best predictor of suicide is prior SRB and that those prior SRB are also associated with comorbidities such as TBI, depression, SUD, and PTSD, all of which are common in Post-9/11 Veterans [35]. Similar to other studies [29, 45, 46], these data suggest that while pain is an important risk factor for suicidal ideation and attempts, psychological comorbidities play a larger role in the development and/or maintenance of suicidal ideation and attempts than pain alone [45]. While the management of chronic pain is paramount within a treatment program, this study highlights the need to add focus to potentially modifiable psychological comorbidities that can drive suicidal ideation and attempts.

The length of the suicidal process may vary among those with chronic pain and mTBI. Individuals with chronic pain alone may have a gradual progression towards suicidal behavior.

Suicide may be deliberative and carefully planned, developed, ruminated, and then systematically carried out. In contrast, those with mTBI may have sudden suicidal behaviors which appear hastily decided-upon with little or no planning. Increased impulsivity associated with frontal lobe damages in those with mTBI may contribute to the sudden emergence of suicide [47]. Therefore, depending on whether mTBI is present, tailored interventions are warranted. With the presence of chronic suicidality, long term mental health and medical treatment are indicated. Impulsivity that leads to suicidal behaviors may seem more difficult to treat; however, due to increased utilization of healthcare by Veterans with mTBI there are ample opportunities to screen for suicide and intervene. Fortunately, there are numerous clinical interventions shown to significantly decrease suicidal behavior [48].

Both mTBI and pain are associated with mental health conditions and physical disabilities leading to difficulty in activities of daily living. These side effects of chronic pain and suicide-related behaviors are hidden disabilities among Veterans with mTBI. Given the high rates of suicide in this population, ongoing monitoring for psychological complications and distress are needed to prevent fatal suicide events. Veterans with mTBI and pain have increased healthcare encounters; thus, multiple opportunities are present to engage patients and caregivers in screening and prevention. Increased screening for suicide within this population is necessary and recommended.

Research related to suicide among patients living with a history of mTBI living with pain is limited. There is insufficient evidence to guide suicide management among those with mTBI and complex comorbid conditions. Among patients with mTBI, the traumatic brain injury is a long-term risk factor for suicide. The current VA/DoD clinical practice guidelines recommend a comprehensive treatment plan by addressing all physical conditions and mental health symptoms simultaneously [49].

## Limitations

Known SRB events were limited to the ICD-codes documented in VA electronic medical records. These SRB data were limited compared to other SRB databases: the VA Suicide Prevention Applications Network (SPAN) [50] and the Joint Department of Defense (DoD)—Department of Veterans Affairs (VA) Suicide Data Repository (SDR) [51–53]. Thus our findings should be interpreted with cautions, including potential biases due to measurement errors (e.g., under detection of non-fatal and fatal suicide attempts, or timing of suicide events) or unmeasured confounding (e.g., factors that influenced both SRB and SRB predictors). Since we did not have access to the VA SPAN and the DoD/VA SDR databases, nor information regarding factors underlying the SRB reporting attrition, we are unable to assess the magnitudes nor directions of these potential biases. However, these biases could be limited as our results were consistent with known suicide predictors in the current literature. Nevertheless, future studies would benefit from combining data from SPAN, SDR and DoD/VA medical records and examining conditional probabilities of factors associated with the highest mortality.

While derived from a national sample, these findings cannot be extrapolated to all patients and should be used with caution due to the possibility of unmeasured confounding. For the purpose of inferring practical intervention to mitigate the sources underlying differential SRB risk associated with pain phenotypes, we chose clinical covariates in the propensity score model that could be modified during clinical care such as mental health conditions that can be treated.

Another limitation is related to the reporting and documentation of suicide attempts and ideations in the patients' medical records. The numbers captured in this study may be lower than the actual number of suicide ideations and attempts that occur among Veterans with mTBI and pain due to reporting and documenting errors. Additionally, our numbers may be

underrepresented because not all Veterans who experience SRB seek treatment at the VA or seek treatment at all due to stigma [54, 55]. It is possible that the rate of suicidal behavior is actually higher than what we found among individuals with TBI because those not seeking treatment may be at higher suicide risk.

## Conclusion

These data expand on the existing knowledge about the impact of pain intensity, suicidal ideation and attempts, and complex patterns of comorbidities. These findings highlight the need for interdisciplinary care as the effect of the phenotypes disappeared when comorbidities were balanced. Assessing and treating both physical and mental health comorbidities are critical to improving outcomes and mitigating risk for suicide-related behavior.

## Supporting information

**S1 Table. ICD-9-CM diagnostic code definitions among Post-9/11 Veterans.**
(DOCX)

**S2 Table. Binomial logistic regression by complex pain phenotype for suicidal ideation or attempt.**
(DOCX)

**S3 Table. Multinomial logistic regression by complex pain phenotype for suicidal ideation.**
(DOCX)

**S4 Table. Multinomial logistic regression by complex pain phenotype for suicide attempt.**
(DOCX)

**S5 Table. Binomial logistic regression by complex pain phenotype for suicidal ideation or attempt.** Adjusted odds ratios and 95% confidence intervals associated with pain phenotypes derived from logistic regression analyses: unadjusted, adjusting away confounding associated with sociodemographic, military characteristics, mental health (minus prior suicide-related behavior) covariates using IPSW.
(DOCX)

**S6 Table. Multinomial logistic regression by complex pain phenotype for suicidal ideation and attempt.** Adjusted odds ratios and 95% confidence intervals associated with pain phenotypes derived from logistic regression analyses: unadjusted, adjusting away confounding associated with sociodemographic, military characteristics, mental health (minus prior suicide-related behavior) covariates using IPSW.
(DOCX)

## Author Contributions

**Conceptualization:** Kangwon Song, Chen-Pin Wang, Donald D. McGeary, Cindy A. McGeary, Jennifer S. Potter, Carlos A. Jaramillo, Blessen C. Eapen, Mary Jo Pugh.

**Data curation:** Ben J. Brintz, Mary Jo Pugh.

**Formal analysis:** Kangwon Song, Ben J. Brintz, Chen-Pin Wang, Donald D. McGeary, Cindy A. McGeary, Jennifer S. Potter, Carlos A. Jaramillo, Blessen C. Eapen, Mary Jo Pugh.

**Funding acquisition:** Mary Jo Pugh.

**Investigation:** Kangwon Song, Ben J. Brintz, Chen-Pin Wang, Donald D. McGeary, Cindy A. McGeary, Jennifer S. Potter, Carlos A. Jaramillo, Blessen C. Eapen, Mary Jo Pugh.

**Methodology:** Kangwon Song, Ben J. Brintz, Chen-Pin Wang, Donald D. McGeary, Cindy A. McGeary, Jennifer S. Potter, Carlos A. Jaramillo, Blessen C. Eapen, Mary Jo Pugh.

**Project administration:** Kangwon Song, Mary Jo Pugh.

**Resources:** Mary Jo Pugh.

**Software:** Ben J. Brintz, Chen-Pin Wang, Mary Jo Pugh.

**Supervision:** Chen-Pin Wang, Mary Jo Pugh.

**Validation:** Ben J. Brintz, Chen-Pin Wang, Mary Jo Pugh.

**Visualization:** Kangwon Song, Mary Jo Pugh.

**Writing – original draft:** Kangwon Song, Ben J. Brintz, Chen-Pin Wang, Donald D. McGeary, Cindy A. McGeary, Jennifer S. Potter, Carlos A. Jaramillo, Blessen C. Eapen, Mary Jo Pugh.

**Writing – review & editing:** Kangwon Song, Ben J. Brintz, Chen-Pin Wang, Donald D. McGeary, Cindy A. McGeary, Jennifer S. Potter, Carlos A. Jaramillo, Blessen C. Eapen, Mary Jo Pugh.

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
