## [Decision Letter · Decision Letter 0]

7 Mar 2022

PONE-D-21-38828Suicidal ideation and attempt among Post-9/11 Veterans with mild traumatic brain injury and complex painPLOS ONE

Dear Dr. Song,

Thank you for submitting your manuscript to PLOS ONE. After careful consideration, we feel that it has merit but does not fully meet PLOS ONE’s publication criteria as it currently stands. Therefore, we invite you to submit a revised version of the manuscript that addresses the points raised during the review process.

We look forward to receiving your revised manuscript.

Kind regards,

Marco Innamorati

Academic Editor

PLOS ONE

Journal Requirements:

3. Please upload a copy of Figures 2 and 3, to which you refer in your text on page 9. If the figure is no longer to be included as part of the submission please remove all reference to it within the text.

Reviewers' comments:

Reviewer's Responses to Questions

**Comments to the Author**

1. Is the manuscript technically sound, and do the data support the conclusions?

Reviewer #1: Yes

Reviewer #2: Yes

2. Has the statistical analysis been performed appropriately and rigorously? 

Reviewer #1: Yes

Reviewer #2: Yes

3. Have the authors made all data underlying the findings in their manuscript fully available?

Reviewer #1: No

Reviewer #2: No

4. Is the manuscript presented in an intelligible fashion and written in standard English?

Reviewer #1: Yes

Reviewer #2: Yes

5. Review Comments to the Author

Reviewer #1: This is an interesting and rare paper focusing on the association of pain and suicdal behaviour and specifically if pain phenotypes are independently associated with suicidal idation or attempts, or via comoribidites, in a longitudinal retrospective cohort study in 9/11 veterans. The objectives are both sicentifically and clinically relevant, methods well chosen and described, results are well presented, thoroughly discussed, limitations are exhaustively addressed and the conclusions are fully supported by the data.

As this is a very well designed study and the results are reported in a simialry excellently written paper I only have a few renarks:

1. Please make the title more espressive of the actual major findings of the study.

2. In the abstract methods section please describe pain phenotypes and also provide some iformation on the statistical analyses performed.

3. Given the reluctance of psychiatric patients to seek help in part due to stigma but also as a symptom of their psychiatric morbidity, and knowing that the risk of suicidal behaiovur is higher among untreated psychiatric patients (including PTSD patients), this should be mentioned as a limitation.

4. What was the reason for not including completed suicides as an outcome?

5. Why were alcohol or substance use disorders not included as covariates?

Reviewer #2: The authors reported original research results derived from their investigation on suicidal ideation and attempt among Post-9/11 Veterans with mild traumatic brain injury and complex pain. The article is overall well-written and of interest to the journal. However, it is unclear how authors assessed suicidal ideation and suicide attempts. Especially for suicidal ideation, the assessment might be more critical. Furthermore, I suggest providing some understanding of the mental pain occurring in the suicidal mind, discussing papers such as Critical appraisal of major depression with suicidal ideation. Ann Gen Psychiatry. 2019

6. PLOS authors have the option to publish the peer review history of their article (what does this mean?). If published, this will include your full peer review and any attached files.

Reviewer #1: No

Reviewer #2: No

---

## [Author Response · Author response to Decision Letter 0]

17 Mar 2022

Reviewer #1: This is an interesting and rare paper focusing on the association of pain and suicidal behavior and specifically if pain phenotypes are independently associated with suicidal ideation or attempts, or via comorbidities, in a longitudinal retrospective cohort study in 9/11 veterans. The objectives are both scientifically and clinically relevant, methods well-chosen and described, results are well presented, thoroughly discussed, limitations are exhaustively addressed and the conclusions are fully supported by the data. As this is a very well designed study and the results are reported in a similarly excellently written paper I only have a few remarks:

1. Please make the title more expressive of the actual major findings of the study. We edited the title per reviewer #1’s recommendations.

2. In the abstract methods section please describe pain phenotypes and also provide some information on the statistical analyses performed. We updated the abstract per reviewer #1’s recommendations.

3. Given the reluctance of psychiatric patients to seek help in part due to stigma but also as a symptom of their psychiatric morbidity, and knowing that the risk of suicidal behavior is higher among untreated psychiatric patients (including PTSD patients), this should be mentioned as a limitation. We updated the limitations and added two references (#53 & 54) to address the reviewer’s comments, see the limitations section, paragraph three, lines 24-28.

4. What was the reason for not including completed suicides as an outcome? At the time the study was completed, we did not have access to data on completed suicides. Thus, our aim was to identify suicidal ideation and suicide attempts that were documented in the medical records.

5. Why were alcohol or substance use disorders not included as covariates? Substance use disorders were included as covariates, see table 1.

Reviewer #2: The authors reported original research results derived from their investigation on suicidal ideation and attempt among Post-9/11 Veterans with mild traumatic brain injury and complex pain. The article is overall well-written and of interest to the journal. 

1. However, it is unclear how authors assessed suicidal ideation and suicide attempts. Especially for suicidal ideation, the assessment might be more critical. We used ICD9-CM codes used in prior studies of SRB to identify suicide ideation (V6284) and attempt (E950, E952, E953, E953, E954, E955, E956, E957, E958, E959) in national VA inpatient and outpatient data in years 6-8 of VA care.

2. Furthermore, I suggest providing some understanding of the mental pain occurring in the suicidal mind, discussing papers such as Critical appraisal of major depression with suicidal ideation. Ann Gen Psychiatry. 2019. We added a sentence to the introduction, paragraph three, lines 22-24 along with the suggested citation (#18).

---

## [Decision Letter · Decision Letter 1]

18 Apr 2022

Complex pain phenotypes: suicidal ideation and attempt through latent multimorbidity

PONE-D-21-38828R1

Dear Dr. Song,

We’re pleased to inform you that your manuscript has been judged scientifically suitable for publication and will be formally accepted for publication once it meets all outstanding technical requirements.

Kind regards,

Marco Innamorati

Academic Editor

PLOS ONE

Additional Editor Comments (optional):

Reviewers' comments:

Reviewer's Responses to Questions

**Comments to the Author**

1. If the authors have adequately addressed your comments raised in a previous round of review and you feel that this manuscript is now acceptable for publication, you may indicate that here to bypass the “Comments to the Author” section, enter your conflict of interest statement in the “Confidential to Editor” section, and submit your "Accept" recommendation.

Reviewer #1: All comments have been addressed

Reviewer #2: All comments have been addressed

2. Is the manuscript technically sound, and do the data support the conclusions?

Reviewer #1: Yes

Reviewer #2: Yes

3. Has the statistical analysis been performed appropriately and rigorously? 

Reviewer #1: Yes

Reviewer #2: Yes

4. Have the authors made all data underlying the findings in their manuscript fully available?

Reviewer #1: Yes

Reviewer #2: Yes

5. Is the manuscript presented in an intelligible fashion and written in standard English?

Reviewer #1: Yes

Reviewer #2: Yes

6. Review Comments to the Author

Reviewer #1: The authors have addressed all comments and recommendations. The paper can be accepted in its present version.

Reviewer #2: The authors addressed my comments and the article appears suitable for possible publication in the journa.

7. PLOS authors have the option to publish the peer review history of their article (what does this mean?). If published, this will include your full peer review and any attached files.

Reviewer #1: No

Reviewer #2: No

---

## [Editor Report · Acceptance letter]

20 Apr 2022

PONE-D-21-38828R1 

Complex pain phenotypes: suicidal ideation and attempt through latent multimorbidity 

Dear Dr. Song:

I'm pleased to inform you that your manuscript has been deemed suitable for publication in PLOS ONE. Congratulations! Your manuscript is now with our production department. 

Kind regards, 

on behalf of

Dr. Marco Innamorati 

Academic Editor

PLOS ONE